# A Green Solution for the Rehabilitation of Marginal Lands: The Case of *Lablab purpureus* (L.) Sweet Grown in Technosols

**DOI:** 10.3390/plants12142682

**Published:** 2023-07-18

**Authors:** Antonio Aguilar-Garrido, Marino Pedro Reyes-Martín, Patrícia Vidigal, Maria Manuela Abreu

**Affiliations:** 1Departamento de Edafología y Química Agrícola, Facultad de Ciencias, Universidad de Granada, Av. de Fuente Nueva s/n, 18071 Granada, Spain; marinoreyes@ugr.es; 2LEAF—Linking Landscape, Environment, Agriculture and Food Research Center, Associate Laboratory TERRA, Instituto Superior de Agronomia (ISA), Universidade de Lisboa, Tapada da Ajuda, 1349-017 Lisboa, Portugal; pvidigal@isa.ulisboa.pt (P.V.); manuelaabreu@isa.ulisboa.pt (M.M.A.)

**Keywords:** environmental rehabilitation, food production, designed Technosols, gossan wastes, phytoremediation, waste valorisation, soil enzymatic activities, potentially hazardous elements, forgotten crops, postmining areas

## Abstract

Reclamation of abandoned mining areas can be a potentially viable solution to tackle three major problems: waste mismanagement, environmental contamination, and growing food demand. This study aims to evaluate the rehabilitation of mining areas into agricultural production areas using integrated biotechnology and combining Technosols with a multipurpose (forage, food, ornamental and medicinal) drought-resistant legume, the *Lablab purpureus* (L.) Sweet. Two Technosols were prepared by combining gossan waste (GW) from an abandoned mining area with a mix of low-cost organic and inorganic materials. Before and after plant growth, several parameters were analysed, such as soil physicochemical characteristics, nutritional status, bioavailable concentrations of potentially hazardous elements (PHE), soil enzymatic activities, and development and accumulation of PHE in *Lablab*, among others. Both Technosols improved physicochemical conditions, nutritional status and microbiological activity, and reduced the bioavailability of most PHE (except As) of GW. *Lablab* thrived in both Technosols and showed PHE accumulation mainly in the roots, with PHE concentrations in the shoots that are safe for cattle and sheep consumption. Thus, this is a potential plant that, in conjunction with Technosols, constitutes a potential integrated biotechnology approach for the conversion of marginal lands, such as abandoned mining areas, into food-production areas.

## 1. Introduction

The global mining industry is growing sharply in response to the increasing demand for strategic elements (e.g., critical metals, rare earth elements, technology-critical elements) needed for energy and digital transition. The large number of operating mines raises health and environmental concerns, many of them related to the large amount of waste generated (over 100 billion tonnes per year [1]). However, the problem is not limited to active mines; the more than one million abandoned mines worldwide [2] after centuries of exploitation also constitute degraded environments. Thus, postmining waste management is necessary to solve a current and future global problem [2], such as the need for a clean-energy transition that relies heavily on mineral resources [3]. The intense extractive activity of sulfide ores generates various types of wastes (gossan waste: gossan materials mixed with host rocks, sulfide waste: crushed pyrite and smelter ash) and leachates; leading to acid mine drainage (AMD) from sulfide materials that can affect soils and surface water and groundwater both in mining areas and for many kilometres around for decades or centuries [4,5].

Thus, this mining can lead to serious consequences for ecosystems (e.g., groundwater contamination, phytotoxicity, and fish mortality) and human health (e.g., nervous system damage, cancers, and mental retardation in children) [6,7,8,9]. A clear example is the Iberian Pyrite Belt, one of the largest metallogenetic provinces of massive sulfides worldwide, with original reserves of more than 1700 Mt [10,11,12], where exists a large legacy of abandoned mines (e.g., São Domingos, Caveira, Aljustrel, San Miguel, and Rio Tinto) and associated tailing dumps (enormous sulfide-bearing waste-rock piles, tailings, and flooded pits); as well as waste produced by operating mines [5,13]. For example, at the Sao Domingos mine, the waste produced was very heterogeneous both in volume/mass produced and in degree of hazardousness. Two main groups of waste were distinguished: mining waste (gossan waste and sulfide country rocks) and industrial waste derived from ore processing operations (slags, iron oxides, smelter ashes, pyrite-rich dumps, leaching tank rejects, and industrial dumps) containing the following quantities of stored potentially hazardous elements (PHE) that could be released into the environment (in tonnes, Fe: 172,514, S: 10,564, Pb: 6644, Zn: 2610, Mn: 1126, Cu: 1032, Cr: 183, As: 109, Sb: 34, and Cd: 0.9) under full exposure conditions, which constitutes a significant source of contamination [14]. Therefore, it is an area with severe environmental problems and could be considered a model among sulfide mines worldwide.

Another major global problem is the increasing global food demand due to population growth. The current population of eight billion people is expected to reach 8.6 billion in 2030, 9.8 billion in 2050 and 11.2 billion in 2100 [15], putting severe pressure on land and water resources; thus, it is imperative to safeguard our natural resources but also to rehabilitate degraded areas. In fact, both society and the scientific community demand a change in food production that allows for lower land use and reduces water consumption and greenhouse-gas emissions [16,17]. In this study, the potential of using green technology towards the rehabilitation of marginal lands, such as abandoned mining areas, is explored. To this end, the combination of waste management allied to the production of a forgotten crop, *Lablab purpureus* (L.) Sweet was studied, evaluating its viability towards the reclamation of marginal lands. In this sense, the combination of Technosols and phytostabilisation techniques has emerged in recent years as one of the most effective technologies for the rehabilitation of degraded areas and their reconversion to agricultural and livestock activities while protecting the food chain [18,19,20,21,22,23].

In particular, Technosols, based on a pedo-engineering, sustainability, and circular economy approach, have been shown to be effective in mining land rehabilitation (soils, tailings, and leachates), both in microcosm studies under controlled conditions [24,25,26,27] and in the field [28,29,30,31]. These consist of human-made soils (called Technosol “a la carte” or tailor-made Technosol [18]) designed specifically for a particular environmental problem, whose properties and pedogenesis are characterised by their technical origin (containing ≥20% artefacts: wastes or materials from anthropic activities [32]). In addition to the general soil functions, Technosols can create/stimulate several biogeochemical and edaphic processes (e.g., acid neutralisation, decrease sulfide oxidation, immobilise PHE, and increase fertility and biotic activity) due to the complementarity of their components [19]. Likewise, phytostabilisation techniques have been widely applied as a low-cost technology for mine rehabilitation in recent decades, in many cases using wild shrub plant species of some economic importance (e.g., *Cistus* sp., *Erica* sp., *Lavandula* sp.), especially for the extraction of valuable compounds with aromatic and medicinal properties. This phytoremediation strategy is based on the fact that many plant species are tolerant to the adverse mine waste or soil conditions (e.g., high total concentrations of PHE, low pH, organic C and nutrients, and poor structure and water-holding capacity); thus, they can naturally colonise mining areas [21,33,34,35,36,37,38], even immobilising PHE in soil or roots, thereby reducing their mobility and bioavailability in the environment [39]. However, seed germination and plant development directly on waste or contaminated soils tends to be slow or even inhibited, especially in harsh climates such as the Mediterranean climate [40,41], limiting the environmental rehabilitation success. To accelerate rehabilitation processes, the combined use of Technosols and plant growth is preferred over only one of these technologies [42]. Since, in general, the addition of different supplementary amendments (organic and inorganic wastes) with specific purposes promotes prolonged biogeochemical and edaphic processes in substrates (e.g., degraded soils, mine wastes) that lead to a decrease in PHE bioavailability and mobility and to an improvement of their physicochemical and biological characteristics favouring plant and microbiota growth [23].

*Lablab purpureus* (L.) Sweet (common name: hyacinth bean or *Lablab*) is a stress-tolerant, versatile, and multipurpose [43] crop native to tropical and subtropical Africa, recognised as one of the crops for the future [44]. It is one of the oldest domesticated and cultivated legumes because of its versatility and multifunctionality (pulse, vegetable [leaf, pod, and flower], forage/green manure, medicinal, phytopharmaceutical, and ornamental) [45]. Furthermore, it is very tolerant to abiotic stress and is highly adaptable to soil conditions. In particular, it is highly resistant to drought via physiological adaptations (e.g., insolation-dependent leaf movement) in low rainfall areas [46] and is moderately tolerant to salinity [47]. This legume has also been shown as a potential phytoremediator of the herbicide trifloxysulfuron sodium [48] and a phytostabiliser of Cd [49]. This suggests that this legume is a potential alternative for the reclamation of abandoned mining areas, especially in harsh climates prone to worsen with climate change, such as in the Iberian Pyrite Belt.

Therefore, the central objective of this study is to evaluate the potential in mining reclamation of green biotechnology that combines the widely tolerant multifunctional legume, *Lablab*, with two designed Technosols derived from mixing gossan wastes with a combination of organic/inorganic wastes from local industries (urban gardening services, quarries, cafes, and breweries). Specifically, the purpose is to promote favourable conditions in the gossan waste for optimal growth of *Lablab* (e.g., improve physicochemical properties and microbiological activity, and decrease PHE bioavailability), as well as to evaluate the safety of this plant for animal consumption.

## 2. Materials and Methods

### 2.1. Experimental Setup

Samples of gossan waste (GW) were collected from the abandoned mining area of São Domingos (37°40′28′′ N, 7°30′01′′ W), in the southeast (SE) of the Iberian Pyrite Belt (Beja, Portugal) (Appendix A), considered as the fourth most hazardous waste produced in this exploitation and the third in terms of volume/mass (with 1.7 hm^3^ and 4.1 Mt) [14]. Thus, the selection of this GW can be considered as a reference model for the study of rehabilitation solutions for sulfide mining areas.

The assay consisted of three treatments (hereafter, substrata) set up in pots of approx. 10 dm^3^ volume. Each treatment had six pots (replicates). The substrata were (i) gossan waste (GW), hereafter GW base (GW*_b_*); (ii) Technosol 50 (TC50) composed of GW mixed with organic and inorganic wastes at 50 g kg^−1^ GW; (iii) Technosol 75 (TC75) consisting of the same mixture of GW and wastes but at a ratio of 75 g kg^−1^ GW. The waste mixture used for the Technosols was constituted of biomass pruning, limestone rock wastes from a quarry (Ø < 5 mm), coffee grounds, sludge, and waste kieselguhr from breweries. This mixture was made manually at 25:20:20:25:10 proportions of the wastes, respectively. A characterisation of the wastes used can be found in [50,51]. All substrata were incubated at 75% of water-holding capacity and room temperature (20–25 °C) for 30 days. This incubation period is considered necessary for the stabilisation of biogeochemical processes in all substrates.

After one month of incubation (starting on 28 January 2019), four seeds of *L. purpureus* (L.) Sweet, cv. Rongai inoculated with *Bradyrhizobium* sp. (strain CB1024) obtained from an Australian seed supplier (Gold Coast Agribusiness Pty. Ltd., Banora Point, NSW, Australia), were sown per pot and substrata. On sowing day (27 February 2019), a composite sample (five subsamples) of the substrata per pot was collected, hereafter referred to as initial GW (GW*_i_*), initial TC50 (TC50*_i_*) and initial TC75 (TC75*_i_*). After 174 days from sowing (19 August 2019), plants (shoot and root) were collected, weighed (fresh and dry weight), and measured (fresh root and shoot). This cultivation period is based on the premise that 130 days after germination *Lablab* is at maximum vegetative growth (corresponding to three trifoliated leaves) [52]; therefore, it is extended to 174 days, when the long stems started to develop, but also to avoid root strangulation in pots. Subsequently, a composite sample (five subsamples) of the substrata per pot was collected, hereafter called, final GW (GW*_f_*), final TC50 (TC50*_f_*), and final TC75 (TC75*_f_*). The complete assay was carried out in a greenhouse under controlled aeration conditions with the substrata kept at 75% of water-holding capacity. After sampling, an aliquot of each soil was stored at 4 °C in sterile, opaque, cold-preserved bottles for the analysis of soil enzymatic activity. The remaining part of the sampled soils was air-dried at room temperature, homogenised, and sieved to 2 mm for physicochemical characterization and determination of multielemental concentrations.

### 2.2. Sample Analysis

Physicochemical characterisation of all substrata (GW*_b_*, GW*_i_*, TC50*_i_*, TC75*_i_,* GW*_f_*, TC50*_f,_* and TC75*_f_*) was performed according to [53]: pH and electrical conductivity (EC) in water suspension (1:2.5 *m*/*V*), total organic C (C_org_) by wet combustion, total N (N_T_) using the Kjeldahl method, extractable P and K (P_Ext_ and K_Ext_) by the Egner–Riehm method, and cation exchange capacity (CEC) by ammonium acetate method for GW*_b_*. Total concentrations of macro- (Ca, Mg, Na, K) and micronutrients (Fe, Mn, Zn and Cu) in GW*_i_*, TC50*_i_*, TC75*_i_,* GW*_f_*, TC50*_f_*, and TC75*_f_* were also analysed by flame atomic absorption spectroscopy after extraction by the Lakanen and Erviö method [54]. Furthermore, it was determined the water-holding capacity by the percolation method at the beginning (GW*_i_*, TC50*_i_*, and TC75*_i_*).

Multielemental (pseudo-total) concentrations of GW*_b_* were determined by instrumental neutron activation analysis and inductively coupled plasma after aqua regia partial digestion (HCl:HNO_3_ 3:1) [55]. Multielemental concentrations in the bioavailable fraction of GW*_i_*, TC50*_i_*, TC75*_i_*, GW*_f_*, TC50*_f,_* and TC75*_f_* were measured by inductively coupled plasma mass spectrometry (ICP-MS) [56,57] after extraction by the rhizosphere-based method [58], which consist of mixing 3 g of moist rhizosphere soil with 20 mL of combined organic acid solution of acetic, lactic, citric, malic, and formic acids.

Plant shoot and root samples were washed with tap water followed by distilled water; roots were also sonicated (after washing) in distilled water in an ultrasound bath for 30 min. Shoots and roots were dried at 40 °C, weighed for dry weight and finally homogenised into a fine powder. A multielemental chemical analysis of shoot and root samples from each substrata was carried out by ICP-MS after digestion (HNO_3_) [56,57].

Activation Laboratories Lda. is a certified laboratory [59]; thus, quality control of the multielemental analyses of substrata and plant samples was performed by Activation Laboratories standards. Quality control of the remaining analyses was carried out by technical replicates, the use of certified standard solutions, and method reagent blanks.

### 2.3. Enzyme Assays

Several enzymatic activities (total fraction) were analysed on the substrata (GW*_i_*, TC50*_i_*, TC75*_i_*, GW*_f_*, TC50*_f_*, TC75*_f_*), as biological indicators to assess the rehabilitation process; namely, dehydrogenase [60] was used as an index of overall microbial activity [61,62] while β-glucosidase [63], acid phosphatase [64], and urease [65] are associated with C, P, and N cycles [66,67,68], respectively. Cellulase was also determined according to [69], which is linked to the C-cycle [70]; and protease activity [71], which includes several enzymes that catalyse the hydrolysis of proteins and oligopeptides to amino acids involved in the N-cycle [72].

### 2.4. Data Analysis

Data were analysed by a one-way ANOVA and Tukey’s (*p* < 0.05) post test performed using GraphPad Prism version 5.00 for Windows (GraphPad Software, San Diego, CA, USA). To evaluate the plant behaviour under the aforementioned substrata growth conditions, the following coefficients were calculated:(i) Biological absorption coefficient = [Root (x)]_(y)_/[elements in bioavailable fraction (x)]_(y)_
(ii) Translocation coefficient = [shoots (x)]_(y)_/[roots (x)]_(y)_
(iii) Soil-plant transfer coefficient = [shoots (x)]_(y)_/[elements in bioavailable fraction (x)]_(y)_
where (x) corresponds to the concentration of a specific element present in (y) that corresponds to GW*_f_*, TC50*_f_*, or TC75*f*.

Plants with a translocation coefficient lower than one are considered nonaccumulators [73,74], while the soil–plant transfer coefficient is higher than one and with a lack of phytotoxicity signs, indicates the level of plant tolerance for PHE [75].

## 3. Results and Discussion

### 3.1. Physicochemical Characterisation of the Gossan Wastes

Physicochemical characteristics of the gossan waste (GW*_b_*) from the São Domingos mining area show that these materials were highly acidic (pH ~ 3.4) and poorly fertile, manifested by a low cation-exchange capacity (CEC < 2.5 cmol_(+)_ kg^−1^), very low concentration of organic C (5.1 g kg^−1^), as well as total N (0.19 g kg^−1^), extractable P (0.003 mg kg^−1^), and K (7.3 mg kg^−1^) (Table 1) [76].

After incubation (GW*_i_*), gossan wastes showed, overall, no significant differences in physicochemical properties from GW*_b_* (Table 1), except for EC, which showed an increase of around 36%. Slight increases were also observed in some properties, such as pH and total N, while others such as organic C and extractable P and K decreased slightly during the incubation period. *Lablab* development (GW*_f_*) had no great influence on GW*_i_* improvement, as no statistically significant changes were found between these two. The high ratio of Ca/Mg both in GW*_i_* and GW*_f_* (>8.0) indicates that these substrata were very unfavourable for plant nutrition [76]. However, most macro- and micronutrient concentrations in gossan waste suffered, overall, an increase from GW*_i_* to GW*_f_* (Table 2). Although without statistically significant differences, except for Mn, there was an increase of 35% in Fe and 39% in Ca, an effect observed with other legume crops [77,78].

The gossan wastes had high total concentrations of several PHE (in g kg^−1^; As: 9.13, Cu: 0.22, Hg: 0.03, Pb: 29.63; Table 3). Thus, these gossan wastes can be considered contaminated with As, Cu, Hg, and Pb for agricultural, residential/parkland, commercial, and industrial uses in most cases, as the concentrations exceeded 830, 3.5, 169, and 658 times, respectively, the most restrictive regulatory level (usually agricultural use) [79,80]. On the other hand, PHE concentrations in the bioavailable fraction of GW*_i_* and GW*_f_* were low, ranging between 0.0034% for As and 9.09% for Cd of the total concentrations in GW*_b_*. Furthermore, PHE bioavailability did not change significantly between GW*_i_* and GW*_f_*, although there were slight changes such as a decrease of about 40% of bioavailable Pb in GW*_f_* (Table 3), probably due to Pb uptake by *Lablab* roots.

The gossan wastes from the São Domingos mine area constitute degraded environments characterised by very acidic pH, low CEC, low nutrient availability, poor organic C, and high total concentrations of several PHEs, like the vast majority of gossan waste from sulfide mining areas. Therefore, it is necessary to neutralise the acid pH, increase nutrient availability and organic C content, and improve structure and water-holding capacity, which are essential characteristics in the rehabilitation of Mediterranean mining areas [81]. For this purpose, the addition of amendments has been widely used, as they can also provide the ability to immobilise PHE by various chemical processes [82,83], although some of these effects may not last over time [18,20]. Therefore, it might be more efficient to produce specific Technosols adjusted to the conditions of each mining waste by mixing amendments for a better promotion and maintenance of biogeochemical processes and a better decrease of bioavailability of PHE for plants [18,20,84]. In this case, a mixture of organic wastes with different organic C quality and stability (biomass pruning, coffee grounds, sludge, and waste kieselguhr) was used to improve fertility in Technosols, as they have high contents of organic C, total N, extractable P and K, and N-NH_4_^+^, as well as to promote soil aggregate formation to improve soil structure, together with limestone rock wastes to provide acidity buffering capacity [50,51]. Although this was not the only criterion for their selection in the construction of Technosols, the fact that they all come from common activities around the world in urban, mining, and agro-industrial environments was also considered; so, it is assumed that they are readily available in other regions affected by the mining industry.

### 3.2. Physicochemical Characterisation of the Technosols

The technology behind Technosols offered a significant improvement to the physicochemical properties and nutritional status (Table 1 and Table 2), demonstrating the beneficial influence that the Technosols approach has in the improvement of gossan wastes. Regardless of the ratio of the GW–organic and inorganic wastes mixture (TC50*_i_* and TC75*_i_*), both Technosols exhibited significant increases in all the parameters studied. The pH raised from an acidic pH in GW*_i_* to a near neutral pH in Technosols and EC increased 1.7 and 2.6 times in TC50*_i_* and TC75*_i_*, respectively (Table 1). Organic C and total N increased by 77% in TC50*_i_* and in TC75*_i_* increased by 83% but, it was extractable P and K that showed the highest significant increase, over 96% (Table 1). Similarly, the concentration of extractable macro- and micro-nutrients significantly increased by more than 66% in both TC50*_i_* and TC75*_i_* because of the organic and inorganic wastes mixtures added, as observed in other studies [85] but TC75*_i_* was more favourable than in TC50*_i_*.

The concentrations of PHE in the bioavailable fraction of Technosols, in comparison with the total concentrations in GW*_b_* (Table 3), showed a decrease of more than 81% (Mn in TC75*_i_*), with some of the PHE below the detection limit (<0.01 mg kg^−1^), such as Hg and Cd. The bioavailability of PHE in Technosols compared with GW*_i_* has undergone element-dependent variations, mainly driven by the increase in organic C and/or pH [86,87], although these variations between Technosols were negligible. In both Technosols, As increased in the bioavailable fraction while Pb decreased, which is consistent with previous studies [9,88,89], where an increase in organic C can both result in reduced mobility/bioavailability of Pb and increased bioavailability of other PHE (e.g., As). Moreover, when pH was higher than 6.5 in non- or low-carbonated soils such as these Technosols, As bioavailability increases, as it can be desorbed from iron oxides and/or organic matter [90,91,92]. The increase of As bioavailability can also be attributed to the increase of extractable P concentration (Table 1) in Technosols [9], as phosphate anions can compete with As anions for binding sites in soil components, increasing As bioavailability [93].

Throughout the *Lablab* vegetative growth (TC50*_f_* and TC75*_f_*), the physicochemical characteristics, nutrient content, and PHE concentration in their bioavailable fraction have experienced variations in both Technosols (Table 1, Table 2 and Table 3). A decrease of 8% in pH from TC50*_i_* to TC50*_f_* was observed and of 10% from TC75*_i_* to TC75*_f_*. Also, EC decreased with plant development in both Technosols, with a decrease of 29% in TC50*_f_* and 13% in TC75*_f_* (Table 1). Contrary to what is expected, since *Lablab* is a legume crop [94], organic C and total N decreased less than 6% with the development of the crop in both Technosols, except for TC75*_f_*, which showed an increase of 7% in total N, which could be explained by the absence of root nodulation inhibited by the presence of PHE [95]. However, there was a significant increase in extractable P content (12% for TC75*_f_* and 18% for TC50*_f_*), and a slight nonsignificant increase in K content (6% for TC75*_f_* and 3% for TC50*_f_*), which was probably due to the release of these elements by organic matter mineralization and mineral weathering (Table 1). Changes in other macro-and micronutrient content during plant growth were much more variable between Technosols. In TC50*_f_*, a significant increase in Ca, Mg, and Mn concentration occurred, while Na, K, Fe, Zn, and Cu concentrations decreased (Table 2). On the contrary, there was an increase in the concentration of all nutrients in TC75*_f_*, except for Cu which decreased and of Zn which remained unchanged. These results contrast to those obtained in [42], which reported a decrease in nutrient concentration with *Cistus ladanifer* L. growth in similar gossan wastes amended by a mixture of inorganic and organic wastes. The development of *Lablab* in both Technosols made no significant differences in PHE bioavailability; however, in TC50*_f_*, there was an overall increase in most of the PHE measured, except for Hg and Pb. Conversely, *Lablab* in TC75*_f_* resulted in an overall decrease for most of PHE, except for Cd, Cu, and Mn (Table 3). These differences between TC50*f* and TC75*f* in PHE bioavailability could be the result of differences in physicochemical characteristics as a consequence of the increased dosage of waste mixture between the two Technosols [96].

### 3.3. Biological Characterisation of the Substrata

The rehabilitation effectiveness of the Technosols containing gossan waste was also assessed by measuring soil enzymatic activities as biological indicators, which reflect soil functional diversity, changes in microbial community composition, and microbial status [62,97]. Soil-enzyme activity is influenced by soil characteristics related to nutrient availability, soil microbial activity, and land use management processes that modified the potential soil-enzyme-mediated substrate catalysis [98]. The most important soil enzymes involved in the C, N, and P cycles in soil are dehydrogenase, β-glucosidase, cellulase, protease, urease, and phosphatase [99,100,101]. Both cellulase and β-glucosidase break down labile cellulose and related carbohydrates, facilitating the activities of other enzymes such as protease and phosphatase [100,102,103]. Proteases are involved in the first phase of N mineralization while urease regulates the release of N-NH_4_^+^ by urea hydrolysis. Phosphatase regulates the hydrolysis of O-P bonds, releasing phosphate from organic matter [101].

Overall, GW*_i_* and GW*_f_* showed low enzymatic activities, whilst TC75*_i_* and TC75*_f_* exhibited higher enzymatic activities (Figure 1). However, at the end of the trial, most of the enzymatic activities decreased. The higher enzymatic activities found in the Technosols are an indication of the good performance of the overall microbial communities involved in the organic matter degradation, mineralization processes, and nutrient cycling [104]. The enzyme activities determined are involved in nutrient cycling, such as C (β-glucosidase and cellulase), N (protease, urease), and P (acid phosphatase).

Dehydrogenase activity often serves as an indicator of the microbiological redox system and microbial oxidative activities in soils [105], but its activity can be significantly affected by several environmental factors, such as soil moisture, oxygen availability, oxidation-reduction potential, pH, organic matter content, soil profile depth, temperature, season, metal contamination, and soil fertilization or pesticides [106]. Thus, explaining the low dehydrogenase activity observed in both GW*_i_* and GW*_f_*, had (<1 µg TPF g^−1^ 16 h^−1^). In comparison, both Technosols showed higher dehydrogenase activity, at TC50*_i_* about 70 µg TPF g^−1^ 16 h^−1^ and at TC75*_i_* approximately 140 µg TPF g^−1^ 16 h^−1^, almost twice as high as TC50*_i_* (Figure 1). These results are consistent with those reported previously in [42,107,108], indicating that the soil microorganism’s activity, evaluated by dehydrogenase activity, is stimulated with the addition of both organic and inorganic amendments to inhospitable materials (e.g., contaminated soil or gossan waste), regardless of the presence or absence of plants. However, in contrast with other species such as *C. ladanifer* [42] or *Triticum aestivum* L. [109], which increased dehydrogenase activity, *Lablab* negatively affected the dehydrogenase activity in both Technosols, especially in TC75*_f_*. In a study with postagricultural forest soils [110], dehydrogenase activity was related to the soil’s organic carbon content, possibly explaining the decrease observed in both final Technosols, as a slight decrease in organic carbon was observed in TC50*_f_* and TC75*_f_*.

Regarding C-cycle-related activities, the same trend of enzymatic activity stimulation was observed in the Technosols. Cellulase showed similar values in both Technosols (µmol glucose g^−1^ 16 h^−1^; TC50: 1.38, TC75: 1.58), without temporal changes, and was higher compared to the GW control (0.07 µmol glucose g^−1^ 16 h^−1^). The β-glucosidase activity after *Lablab* growth was similar in both Technosols (TC50*_f_* and TC75*_f_*: ~ 0.5 μmol ρ-nitrophenol g^−1^ h^−1^) but, in TC50*_i_*, it is half the activity given in TC75*_i_* (0.37 vs. 0.75 μmol ρ-nitrophenol g^−1^ h^−1^). Thus, in TC50*_f_*, it increased nonsignificantly and, on the contrary, in TC75*_f_* there was a significant decrease.

Unlike the other enzyme activities here studied, GW showed acid phosphatase activity (0.57 μmol ρ-nitrophenol g^−1^ h^−1^), although lower than in TC50 and much lower than in TC75 (0.74 and 1.34 μmol ρ-nitrophenol g^−1^ h^−1^, respectively). At the end of plant development, phosphatase activity was only modified at TC75 where it decreased significantly. Thus, the obtained results are in line with many studies that show a significant increase in organic C by the application of amendments in GW or the construction of tailor-made Technosols, which have a positive effect not only on the dehydrogenase activity but also on the enzymatic activities related to C-cycle (β-glucosidase and cellulase) and P-cycle (acid phosphatase) [42,111,112].

The behaviour of enzymes related to the N-cycle (protease and urease) was similar. The lowest values were found in GW (0.14 μmol tyrosine g^−1^ 16 h^−1^ and 0.18 μmol N-NH_4_ g^−1^ 2 h^−1^, respectively), while in TC50*_i_* were found intermediate values (5.16 μmol tyrosine g^−1^ 16 h^−1^ and 2.92 μmol N-NH_4_ g^−1^ 2 h^−1^, respectively); and in TC75*_i_* maximum activity occurred (7.72 μmol tyrosine g^−1^ 16h^−1^ and 6.91 μmol N-NH_4_ g^−1^ 2h^−1^, respectively). This is related to the higher concentration of total N in Technosols than in GW (0.30 g kg^−1^) and TC75 (1.80 g kg^−1^) compared to TC50 (1.10 g kg^−1^) as high activity, for example of urease, is related to a higher fraction of plant available N [113]. Similar results of N-cycle enzyme activity (mainly protease) have been detailed for metal- and semimetal-contaminated soils amended with a wide variety of wastes, e.g., sewage sludge [114], effluents from cotton ginning mills [115], and pig slurry [116]. In the present study, the urease activity was reduced in both Technosols at the end of the trial, while the protease enzyme was not. The activity of urease is closely related to the biological cycle transformation and the bioavailability of N, whilst proteolysis is a process in the N cycle and is considered as a stage limiting the rate of N mineralization in soils. Thus, the increase in total N TC75*_f_* could be linked to the activity of protease that did not suffer great variation, rather than urease.

### 3.4. Ecotoxicological Characterisation of the Substrata

To assess the ecotoxicological risk of the rehabilitated materials (gossan wastes) by Technosols, a bioassay with *Lablab* has been carried out. In this way, the growing conditions on these materials can also be examined to diagnose the suitability of the combined use of Technosols and this wide-tolerant multifunctional legume to reconvert degraded areas into productive ones.

The germination percentage of *Lablab* was greater in Technosols (TC50: 50%, TC75: 75%) when compared to GW (38%). This increase in germination can be explained, as [9] reported for other plant species (*Lavandula pedunculata* (Mill.) Cav. and *C. ladanifer*), by an improvement in water-holding capacity in the studied Technosols, which is related to an enhancement in structure, due to the wastes used in their preparation. Likewise, the development of *Lablab*, measured in length and dry biomass, has been superior to that of GW, especially in dry biomass (Figure 2 and Figure 3). The shoot length of *Lablab* developed on both Technosols was about 20 times greater than in GW. As for root length, greater variability was observed than in shoot length but also with much more evident growth in Technosols than in GW. Although *Lablab* seeds came inoculated with *Bradyrhizobium* sp. (strain CB1024) from the seed supplier, nodulation failed to occur, as previously reported in other studies [117,118]. The greatest differences between the substrata were observed in the plant biomass. Growth stimulation of these plants is evident in both Technosols, with high biomass production in both shoot and root while, in GW, it is practically nil. Similar results have been reported for several plant species (e.g., *Eucalyptus globulus* Labill., *C. ladanifer*, *Dactylis glomerata* L., *Erica australis* L., and pasture species) grown in different Technosols also composed of mining waste/contaminated soils, at medium–long term, either in field or greenhouse assays [20,42,112]. The plants’ growth can be explained by the improvement of some substrata properties such as those mentioned above, which improve seed germination but, above all, by the increase in nutrients concentration and organic C, as well as biological activity [9]. No statistically significant variations were obtained in the analysed growth parameters of *Lablab* between TC50 and TC75.

The concentrations of PHE and nutrients in plant tissues of *Lablab* (shoots and roots) were determined to test whether this legume can uptake them from soil to roots and then translocate them to shoots; thus, determining the potential use of *Lablab* in phytostabilisation of gossan wastes and as a crop for feed production. In general, PHE concentrations in shoots and roots were higher in plants grown on GW than in those grown on Technosols (Table 4), which is reflected in the higher biological absorption and soil–plant transfer coefficients (Table 5). While plant uptake and accumulation of nutrients in shoots and roots was more variable depending on the nutrient, which may be due to the physiology of this species, some nutrients were at higher concentrations (Ca, K, and Mn) in plants from Technosols and others in lower (Cu, Zn, Fe, and Na) or similar (Mg). However, independently of the studied Technosol, no significant differences were obtained among concentrations, both in shoots and roots, of the nutrients studied in *Lablab* plants. The concentration of Fe in plant shoots grown in GW exceeded by far normal values in plants (50–250 mg kg^−1^ [119]), in TC75 slightly, and in TC50 it was within these. In both GW and Technosols plants, Ca and Mg were found in concentrations within the normal range (Ca: 2–40 g kg^−1^, Mg: 1–8 g kg^−1^ [119]); however, K was in deficit concentrations in GW plants (<20–25 g kg^−1^ [119]). Even some elements (Cu and Zn), which have a micronutrient–PHE duality depending on the concentration, were in the optimal range for good plant development (Cu: 5–20 mg kg^−1^, Zn: 25–150 mg kg^−1^ [119]) in both GW and Technosols plants.

Concentrations of most PHEs studied in shoots of *Lablab* growing in Technosols (Table 4) were considered as normal/sufficient and/or below the phytotoxicity level, except for As, which accumulated in shoots in concentrations that may be toxic to plants (5–20 mg kg^−1^ [86]), although they showed no signs of toxicity (Figure 2). However, in the shoots of *Lablab* growth in the gossan waste, the concentrations of As (168 mg kg^−1^) and Pb (498 mg kg^−1^) were well above the maximum limit considered phytotoxic for plants in general (Pb: 30–300 mg kg^−1^ [86]) and Cr and Ni were close to the limits (5 and 10 mg kg^−1^ [86]), respectively. The high concentrations of these elements in shoots and roots can be responsible for the growth impairment of the plants on GW (Figure 2 and Figure 3).

Most PHE and nutrient concentrations except Mn, Ca, Mg, and K in roots were much higher than in shoots, both in GW and Technosols (Table 4). Moreover, except for Mn, the translocation coefficient was less than a unit in all cases (Table 5), indicating a low transfer of the elements between the root and the aerial part. The calculated soil–plant coefficient transfer (Table 5) had values for plants from GW much higher than those of plants growing in the Technosols. Although these values were also higher than a unit, indicating a high level of tolerance of this legume to PHE, they showed no signs of toxicity, in agreement with several studies [120,121,122]. Thus, *Lablab* could have mechanisms of tolerance to high concentrations of PHE in the soil, as has been observed in many other species of *L. pedunculata* and *C. ladanifer* [9]. Therefore, it could be used in phytostabilisation programmes for gossan wastes. Since it is a plant capable of absorbing PHEs in their bioavailable form and concentrating them in the roots (biological absorption coefficient > 1, translocation coefficient < 1, respectively; Table 5), the combined use of this species with Technosols is a good option as the organic and inorganic wastes composing the Technosols decreased the bioavailability of most PHE in soil and, therefore, the potential uptake by the plant (Table 4).

On the other hand, to verify whether this strategy of converting degraded areas into production areas by combining Technosols and *Lablab* growth is valid, the safety of the consumption of this plant by animals must be evaluated. For this purpose, the concentrations of PHE in shoots were compared with the maximum tolerable levels of PHE in feed for different species (rodents, poultry, pigs, horses, cattle, and sheep) [123]. In this sense, although the concentrations of As in *Lablab* shoots grown in Technosols exceed the phytotoxicity limit, this would not represent a tolerable maximum level of this element (30 mg kg^−1^ [123]) in the diet of the animals aforementioned. However, the maximum tolerable level of Pb in *Lablab* grown in Technosols was exceeded for rodents, poultry, pigs, and horses (10 mg kg^−1^), but not for cattle or sheep feed (100 mg kg^−1^) [123]. In the case of plants grown on gossan waste, there is a risk of As and Pb affection in case of use in feed production for all animal species considered, by Hg for rodents, poultry, pigs and horses feed, and by Cu for sheep feed; as their concentrations in *Lablab* shoots exceeded 30, 10, 0.2, and 15 mg kg^−1^, respectively (most restrictive maximum tolerable level in feed [123]). All other PHEs considered were in *Lablab* shoots at concentrations below the maximum tolerable level in feed for the considered species.

## 4. Conclusions

The gossan wastes present in the abandoned mine area of São Domingos constitute a degraded environment characterised by very low pH values, low CEC, low nutrients availability, poor organic matter, high total concentrations of several PHEs, and low microbiological activity, like the vast majority of gossan wastes from sulfide mining areas. Therefore, this case could be a reference model for the study of rehabilitation solutions for sulfide mining areas, particularly in harsh climates prone to worsen with climate change, such as the Mediterranean region, like the Iberian Pyrite Belt.

The integrated green biotechnology approach combined the rehabilitation of sulfide mine areas by the use of designed Technosols constructed from mixing gossan wastes with a combination of organic/inorganic wastes from local industries (urban gardening services, quarries, cafes, and breweries), followed by the development of *Lablab*, is efficient over the time span tested (174 days) under greenhouse conditions. Both Technosols improved the physicochemical conditions, nutrient status, and microbiological activity of the gossan waste and reduced the bioavailability of most PHEs (with the exception of As); although, in general, conditions in TC75 were more favourable than in TC50. This, in turn, allowed *Lablab* growth with no visible signs of toxicity and low translocation of PHE to the shoots, with PHE concentrations suitable for cattle or sheep but not for poultry, pig, and horse due to the Pb concentration. Whereas in pure gossan wastes, not only was the growth of *Lablab* limited (very low biomass and signs of toxicity), but As, Cu, Hg and Pb concentrations in plants exceeded the limits for most animal feed. Therefore, the combination of Technosols with the *Lablab* crop is a potentially valuable green biotechnology approach for the rehabilitation of degraded/contaminated environments, but also for converting them into food production areas under the condition of exhaustive quality and food-safety controls. Furthermore, in the case of Technosols implementation in mining areas, monitoring should be carried out to assess over time whether conditions are maintained or, for example, whether the addition (or not) of certain amendments to the Technosols needs to be readjusted.

## Figures and Tables

**Figure 1 plants-12-02682-f001:**
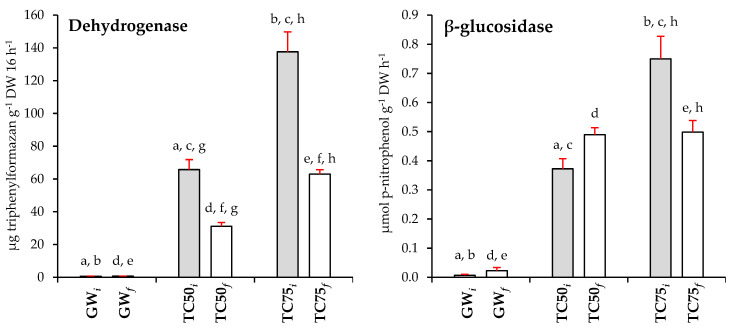
Enzymatic activities (dehydrogenase, β-glucosidase, acid phosphatase, cellulase, urease, and protease) in the gossan wastes and Technosols before (initial) and after (final) *Lablab* growth. GW*_i_*—Initial gossan waste; GW*_f_*—Final gossan waste; TC50*_i_*—Initial Technosol 50 g kg^−1^; TC50*_f_*—Final Technosol 50 g kg^−1^; TC75*_i_*—Initial Technosol 75 g kg^−1^; TC75*_f_*—Final Technosol 75 g kg^−1^; DW—dry weight. Values correspond to the average of six biological replicates (±SE) followed by different letters indicating significant differences between substrata (*p* < 0.05): (a)—GW*_i_* vs. TC50*_i_*, (b)—GW*_i_* vs. TC75*_i_*, (c)—TC50*_i_* vs. TC75*_i_*, (d)—GW*_f_* vs. TC50*_f_*, (e)—GW*_f_* vs. TC75*_f_*, (f)—TC50*_f_* vs. TC75*_f_*, (g)—TC50*_i_* vs. TC50*_f_*; (h)—TC75*_i_* vs. TC75*_f_*.

**Figure 2 plants-12-02682-f002:**
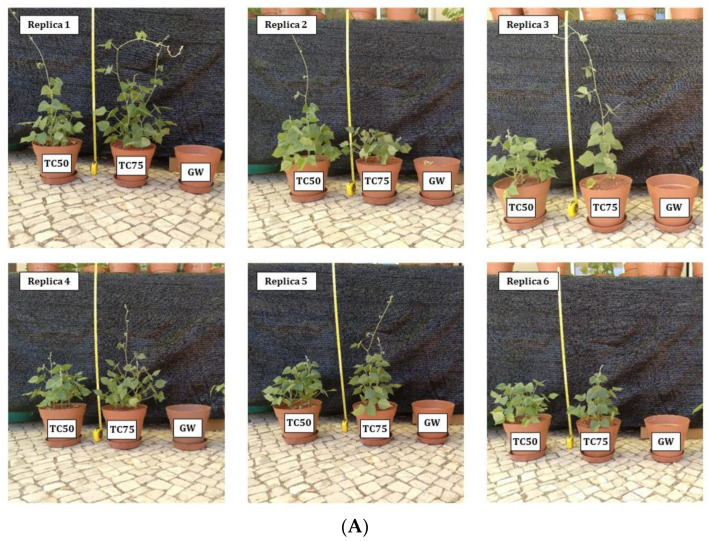
Aspect of *Lablab* plants after six months of growth in the gossan wastes and Technosols: General view (**A**) and details of the roots and aerial part of the plants (**B**). TC50—Technosol 50 g kg^−1^; TC75—Technosol 75 g kg^−1^; GW—Gossan Waste.

**Figure 3 plants-12-02682-f003:**
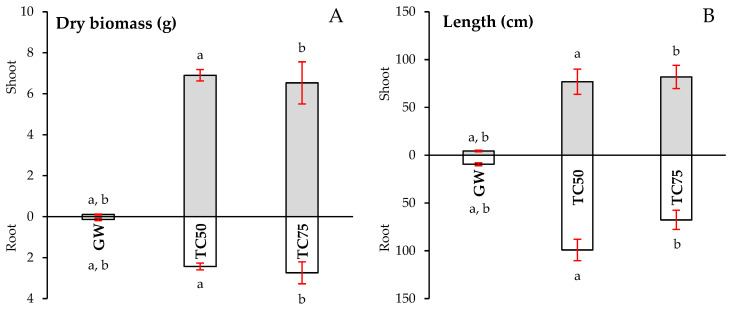
Dry biomass (**A**) and length (**B**) of *Lablab* plants after six months of growth in the gossan wastes and Technosols. GW—Gossan Waste; TC50—Technosol 50 g kg^−1^; TC75—Technosol 75 g kg^−1^. Values correspond to the average of six biological replicates (± SE) followed by different letters indicating significant differences between substrata (*p* < 0.05): (a)—GW vs. TC50; (b)—GW vs. TC75.

**Table 1 plants-12-02682-t001:** Physicochemical characteristics of the gossan wastes and Technosols.

	GW*_b_*	GW*_i_*	TC50*_i_*	TC75*_i_*	GW*_f_*	TC50*_f_*	TC75*_f_*
pH (H_2_O)	3.41 ± 0.01 ^b,c^	3.58 ± 0.09 ^d,e^	6.79 ± 0.03 ^b,d,f,j^	7.04 ± 0.02 ^c,e,f,k^	3.73 ± 0.04 ^g,h^	6.26 ± 0.02 ^g,j^	6.36 ± 0.02 ^h,k^
EC (mS cm^−1^)	0.14 ± 0.02 ^a,b,c^	0.43 ± 0.03 ^d,e^	0.75 ± 0.06 ^d,f,j^	1.12 ± 0.05 ^e,f^	0.29 ± 0.06 ^g,h^	0.54 ± 0.06 ^g,i,j^	0.97 ± 0.03 ^h,i^
C_org_ (g kg^−1^)	5.14 ± 1.18 ^b,c^	3.46 ± 0.46 ^d,e^	15.07 ± 2.01 ^b,d^	20.32 ± 1.30 ^c,e^	4.22 ± 0.17 ^g,h^	14.36 ± 0.86 ^g^	19.71 ± 1.90 ^h^
N_T_ (g kg^−1^)	0.19 ± 0.01 ^b,c^	0.30 ± 0.03 ^d,e^	1.12 ± 0.06 ^b,d,f^	1.82 ± 0.14 ^c,e,f^	0.19 ± 0.04 ^g,h^	1.06 ± 0.04 ^g,i^	1.94 ± 0.07 ^h,i^
P_Ext_ (mg kg^−1^)	0.003 ± 0.001 ^b,c^	n.d. ^d,e^	161.66 ± 16.10 ^b,d,f,j^	358.33 ± 15.31 ^e,f^	0.75 ± 0.41 ^g,h^	180.59 ± 9.50 ^g,i,j^	421.38 ± 30.99 ^h,i^
K_Ext_ (mg kg^−1^)	7.31 ± 3.37 ^b,c^	6.50 ± 2.86 ^d,e^	156.45 ± 7.47 ^b,d,f^	276.15 ± 16.88 ^e,f,k^	6.70 ± 2.41 ^g,h^	161.52 ± 6.93 ^g,i^	292.90 ± 11.20 ^h,i,k^
CEC (cmol_+_ kg^−1^)	2.44 ± 0.49	n.a.	n.a.	n.a.	n.a.	n.a.	n.a.

GW*_b_*—Base gossan waste; GW*_i_*—Initial gossan waste; GW*_f_*—Final gossan waste; TC50*_i_*—Initial Technosol 50 g kg^−1^; TC50*_f_*—Final Technosol 50 g kg^−1^; TC75*_i_*—Initial Technosol 75 g kg^−1^; TC75*_f_*—Final Technosol 75 g kg^−1^; EC—Electrical conductivity; C_org_—Organic carbon; N_T_—Total nitrogen; P_Ext_—Extractable phosphorus; K_Ext_—Extractable potassium; CEC—Cation exchange capacity, n.d.—Non-detectable, n.a.—Not analysed. Values correspond to the average of six biological replicates (± SE) followed by different letters indicating significant differences between substrata (*p* < 0.05): (a)—GW*_b_* vs. GW*_i_*, (b)—GW*_b_* vs. TC50*_i_*, (c)—GW*_b_* vs. TC75*_i_*, (d)—GW*_i_* vs. TC50*_i_*, I—GW*_i_* vs. TC75*_i_*, (f)—TC50*_i_* vs. TC75*_i_*, (g)—GW*_f_* vs. TC50*_f_*, (h)—GW*_f_* vs. TC75*_f_*; (i)—TC50*_f_* vs. TC75*_f_*; (j)—TC50*_i_* vs. TC50*_f_*; (k)—TC75*_i_* vs. TC75*_f_*.

**Table 2 plants-12-02682-t002:** Concentration (mg kg^−1^) of extractable macro- (Ca, Mg, Na, K) and micronutrients (Fe, Mn, Zn, and Cu) [54] in the gossan wastes and Technosols before (initial) and after (final) *Lablab* growth.

Element	GW*_i_*	TC50*_i_*	TC75*_i_*	GW*_f_*	TC50*_f_*	TC75*_f_*
Ca	80.63 ± 11.47 ^a,b^	1075.65 ± 96.90 ^a,h^	1501.75 ± 71.08 ^b,i^	205.28 ± 25.53 ^d,e^	3017.38 ± 212.05 ^d,f,h^	4132.25 ± 337.64 ^e,f,i^
Mg	6.13 ± 0.43 ^a,b^	52.25 ± 2.95 ^a,c,h^	85.20 ± 4.20 ^b,c,i^	8.40 ± 1.02 ^d,e^	66.50 ± 1.21 ^d,f,h^	109.96 ± 5.14 ^e,f,i^
Na	44.25 ± 3.72 ^a,b^	131.90 ± 34.95 ^a^	162.40 ± 6.75 ^b^	41.05 ± 2.63 ^d,e^	104.38 ± 4.41 ^d,f^	178.17 ± 10.53 ^e,f^
K	12.13 ± 1.30 ^a,b^	192.85 ± 11.02 ^a,c^	285.35 ± 12.05 ^b,c^	17.15 ± 2.33 ^d,e^	177.75 ± 7.73 ^d,f^	301.83 ± 13.38 ^e,f^
Fe	24.44 ± 0.70 ^a,b^	1031.35 ± 68.69 ^a,c,h^	1797.95 ± 102.65 ^b,c,i^	69.85 ± 2.14 ^d,e^	850.71 ± 19.99 ^d,f,h^	2024.83 ± 169.79 ^e,f,i^
Mn	<0.01 ^a,b,g^	3.72 ± 0.11 ^a,c,h^	7.03 ± 0.30 ^b,c,i^	1.95 ± 0.34 ^d,e,g^	9.83 ± 0.87 ^d,f,h^	14.63 ± 0.82 ^e,f,i^
Zn	1.31 ± 0.12 ^a,b^	5.22 ± 0.26 ^a,c,h^	7.61 ± 0.77 ^b,c^	1.31 ± 0.21 ^d,e^	3.88 ± 0.07 ^d,f,h^	7.59 ± 0.51 ^e,f^
Cu	6.03 ± 2.05 ^g^	7.10 ± 0.72	5.58 ± 0.32	1.24 ± 0.03 ^g^	3.04 ± 0.08	4.78 ± 0.31

GW*_i_*—Initial gossan waste; GW*_f_*—Final gossan waste; TC50*_i_*—Initial Technosol 50 g kg^−1^; TC50*_f_*—Final Technosol 50 g kg^−1^; TC75*_i_*—Initial Technosol 75 g kg^−1^; TC75*_f_*—Final Technosol 75 g kg^−1^. Values correspond to the average of six biological replicates ± SE followed by different letters indicating significant differences between substrata (*p* < 0.05): (a)—GW*_i_* vs. TC50*_i_*, (b)—GW*_i_* vs. TC75*_i_*, (c)—TC50*_i_* vs. TC75*_i_*, (d)—GW*_f_* vs. TC50*_f_*, (e)—GW*_f_* vs. TC75*_f_*, (f)—TC50*_f_* vs. TC75*_f_*, (g)—GW*_i_* vs. GW*_f_*; (h)—TC50*_i_* vs. TC50*_f_*; (i)—TC75*_i_* vs. TC75*_f_*.

**Table 3 plants-12-02682-t003:** Concentration of potentially hazardous elements in the base gossan wastes (pseudototal) and in the bioavailable fraction of the gossan wastes and Technosols before (initial) and after (final) *Lablab* growth.

	Pseudototal Concentration (mg kg^−1^)	Bioavailable Concentration (mg kg^−1^)					
Elements	GW*_b_*	GW*_i_*	TC50*_i_*	TC75*_i_*	GW*_f_*	TC50*_f_*	TC75*_f_*
As	9126.67 ± 238.77	0.31 ± 0.06 ^a,b^	2.14 ± 0.47 ^a^	2.57 ± 0.38 ^b^	0.32 ± 0.03 ^c^	3.23 ± 0.62 ^c,e^	1.59 ± 0.22 ^e^
Cd	0.11 ± 0.08	0.01 ± 0.00	<0.01	<0.01	0.01 ± 0.00	<0.01	<0.01
Cr	21.00 ± 1.15	0.02 ± 0.00 ^b^	0.07 ± 0.01	0.12 ± 0.01 ^b^	0.01 ± 0.00 ^c,d^	0.10 ± 0.02 ^c^	0.09 ± 0.01 ^d^
Cu	218.67 ± 5.81	0.92 ± 0.11 ^a^	0.48 ± 0.09 ^a^	0.56 ± 0.08	0.89 ± 0.05	0.58 ± 0.10	0.73 ± 0.03
Hg	26.67 ± 6.67	<0.01	<0.01	<0.01	<0.01	<0.01	<0.01
Mn	27.67 ± 1.76	0.36 ± 0.04 ^a,b^	3.40 ± 0.50 ^a^	5.28 ± 0.35 ^b^	0.85 ± 0.15 ^c,d^	4.17 ± 0.78 ^c^	5.73 ± 0.58 ^d^
Ni	2.77 ± 0.43	0.09 ± 0.01	0.17 ± 0.05	0.22 ± 0.02	0.10 ± 0.01	0.19 ± 0.04	0.18 ± 0.03
Pb	29,633.33 ± 554.78	2.64 ± 0.35	0.52 ± 0.36	0.15 ± 0.10	1.61 ± 0.24	0.27 ± 0.12	0.10 ± 0.03
Zn	83.33 ± 6.62	3.57 ± 0.54	2.49 ± 0.69	3.59 ± 0.98	5.07 ± 0.35	2.71 ± 0.71	2.53 ± 0.42

GW*_b_*—Base gossan waste; GW*_i_*—Initial gossan waste; GW*_f_*—Final gossan waste; TC50*_i_*—Initial Technosol 50 g kg^−1^; TC50*_f_*—Final Technosol 50 g kg^−1^; TC75*_i_*—Initial Technosol 75 g kg^−1^; TC75*_f_*—Final Technosol 75 g kg^−1^. Values correspond to the average of six biological replicates (± SE) followed by different letters indicating significant differences between substrata (*p* < 0.05): (a)—GW*_i_* vs. TC50*_i_*, (b)—GW*_i_* vs. TC75*_i_*, (c)—GW*_f_* vs. TC50*_f_*, (d)—GW*_f_* vs. TC75*_f_*, (e)—TC50*_f_* vs. TC75*_f_*,

**Table 4 plants-12-02682-t004:** Total concentration (mg kg^−1^) of potentially hazardous elements and nutrients in shoots and roots of *Lablab* grown in the gossan wastes and Technosols.

	Shoots			Roots		
mg kg^−1^	GW	TC50	TC75	GW	TC50	TC75
As	167.62 ± 66.59 ^d^	7.50 ± 1.87	19.52 ± 11.77 ^f^	3228.30 ± 1116.92 ^d^	1,303.01 ± 92.06	2559.03 ± 440.64 ^f^
Cd	0.13 ± 0.03	0.04 ± 0.00	0.04 ± 0.00	2.47 ± 2.26	0.12 ± 0.01	0.08 ± 0.01
Cr	2.76 ± 0.31	1.51 ± 0.07	1.65 ± 0.13	33.90 ± 13.44	6.69 ± 0.86	64.33 ± 41.15
Cu	22.63 ± 1.90 ^d^	7.63 ± 0.32 ^e^	7.73 ± 1.06 ^f^	120.35 ± 17.51 ^b,c,d^	54.96 ± 3.14 ^b,e^	66.76 ± 0.78 ^c,f^
Hg	0.64 ± 0.15 ^d^	0.05 ± 0.00	0.05 ± 0.01	10.36 ± 4.80 ^d^	2.09 ± 0.07	2.49 ± 0.16
Mn	63.32 ± 25.18	80.22 ± 7.04	82.62 ± 9.48	30.70 ± 3.29	43.48 ± 3.79	50.17 ± 2.08
Ni	7.99 ± 1.65	1.88 ± 0.10	1.46 ± 0.14	19.09 ± 2.50	4.02 ± 0.23	18.17 ± 9.37
Pb	497.79 ± 204.05 ^d^	19.61 ± 4.42	60.14 ± 27.34 ^f^	8268.16 ± 2728.47 ^d^	3510.74 ± 240.93	6841.61 ± 1228.20 ^f^
Zn	77.93 ± 27.08 ^d^	33.85 ± 2.88	28.71 ± 2.81	252.21 ± 44.02 ^b,c,d^	115.95 ± 18.28 ^b^	92.94 ± 5.82 ^c^
Fe	3455.75 ± 1763.80 ^d^	186.60 ± 37.68	443.60 ± 191.42	72,248.52 ± 27,067.29 ^d^	26,385.28 ± 2722.85	43,524.84 ± 4186.81
Ca	14,283.28 ± 6080.72	14,497.45 ± 324.90	20,635.08 ± 1584.08	5131.99 ± 1367.83	9684.34 ± 718.13	14,012.42 ± 1983.44
Mg	2235.75 ± 330.09 ^d^	1828.00 ± 55.37	2241.47 ± 167.64	615.73 ± 138.83 ^b,c,d^	1907.58 ± 118.10 ^b^	1581.21 ± 51.54 ^c^
Na	2729.43 ± 82.55 ^d^	1612.31 ± 41.47	1177.15 ± 274.94	6499.61 ± 1863.28 ^d^	3439.55 ± 366.84	3787.01 ± 437.27
K	16,144.79 ± 1886.70 ^a,d^	20,481.38 ± 505.07	24,214.06 ± 1769.29 ^a,f^	7053.64 ± 2300.41 ^b,d^	18,057.22 ± 2174.36 ^b^	11,592.39 ± 361.87 ^f^

GW—Gossan waste; TC50—Technosol 50 g kg^−1^; TC75—Technosol 75 g kg^−1^. Values correspond to the average of four biological replicates ±SE followed by different letters indicating significant differences between substrata (*p* < 0.05): (a)—GW shoot vs. TC75 shoot, (b)—GW root vs. TC50 root, (c)—GW root vs. TC75 root, (d)—GW shoot vs. GW root, (e)—TC50 shoot vs. TC50 root, (f)—TC75 shoot vs. TC75 root.

**Table 5 plants-12-02682-t005:** Calculated biological absorption, translocation, and soil–plant transfer coefficients in the gossan wastes and Technosols.

	Biological Absorption Coefficient ^(1)^			Translocation Coefficient ^(2)^			Soil–Plant Transfer Coefficient ^(3)^		
	GW	TC50	TC75	GW	TC50	TC75	GW	TC50	TC75
As	10,197.2 ± 3635.5 ^a,b^	465.8 ± 118.0 ^a^	1,795.6 ± 544.6 ^b^	0.09 ± 0.05	0.01 ± 0.00	0.01 ± 0.01	590.3 ± 284.0	2.3 ± 0.3	13.9 ± 8.8
Cd	605.7 ± 576.7	97.5 ± 65.8	29.0 ± 4.1	0.48 ± 0.16	0.30 ± 0.02	0.47 ± 0.02	20.1 ± 4.0	24.8 ± 14.9	13.7 ± 2.2
Cr	2273.9 ± 767.7 ^a^	92.3 ± 36.2 ^a^	648.8 ± 390.0	0.1 ± 0.03 ^a^	0.22 ± 0.02 ^a,c^	0.06 ± 0.02 ^c^	193.7 ± 34.4 ^a,b^	18.9 ± 5.3 ^a^	19.0 ± 4.3 ^b^
Cu	139.7 ± 26.6	106.0 ± 24.3	92.0 ± 5.1	0.20 ± 0.03 ^b^	0.14 ± 0.01	0.12 ± 0.02 ^b^	25.7 ± 2.6 ^a,b^	14.4 ± 2.4 ^a^	10.6 ± 1.4 ^b^
Hg	3171.6 ± 1444.4	647.6 ± 25.2	773.3 ± 45.3	0.09 ±0.04	0.02 ± 0.00	0.02 ± 0.00	196.8 ± 47.6 ^a,b^	14.1 ± 1.0 ^a^	16.2 ± 4.1 ^b^
Mn	39.6 ± 7.6 ^a,b^	12.6 ± 3.9 ^a^	9.1 ± 1.1 ^b^	2.03 ± 0.69	1.85 ± 0.08	1.64 ± 0.16	82.2 ± 36.8	22.7 ± 6.3	15.1 ± 2.6
Ni	207.9 ± 54.1 ^a^	24.8 ± 7.3 ^a^	95.8 ± 43.5	0.45 ± 0.14	0.47 ± 0.04	0.16 ± 0.06	82.3 ± 17.6 ^a,b^	11.5 ± 3.0 ^a^	9.2 ± 2.7 ^b^
Pb	4935.7 ± 1291.6	23,498.4 ± 10,331.7	45,658.8 ± 1378.0	0.11 ± 0.06	0.01 ± 0.00	0.01 ± 0.01	391.0 ± 210.7	115.5 ± 38.2	1095.0 ± 526.2
Zn	52.0 ± 12.6	59.7 ± 24.0	41.7 ± 10.7	0.29 ± 0.05	0.30 ± 0.03	0.31 ± 0.04	16.4 ± 6.8	16.7 ± 5.5	12.4 ± 2.4

GW—Gossan Waste; TC50—Technosol 50 g kg^−1^; TC75—Technosol 75 g kg^−1^. Values correspond to the average of four biological replicates (±SE) followed by different letters indicating significant differences between substrata (*p* < 0.05): (a)—GW vs. TC50; (b)—GW vs. TC75; (c)—TC50 vs. TC75. (1) Biological absorption coefficient = [Root (x)]_(y)_/[elements in bioavailable fraction (x)]_(y)_; (2) Translocation coefficient = [shoots (x)]_(y)_/[roots (x)]_(y)_; (3) Soil–plant transfer coefficient = [shoots (x)]_(y)_/[elements in bioavailable fraction (x)]_(y)_. x—Concentration of element present in y, y—GW*_f_*, TC50*_f_*, TC75*f.*

## Data Availability

The data presented in this study are available on request from the corresponding author.

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
