# Peer review of "A Green Solution for the Rehabilitation of Marginal Lands: The Case of Lablab purpureus (L.) Sweet Grown in Technosols"

_plants, 2023, doi:10.3390/plants12142682_

Round 1

Reviewer 1 Report

The matter of the manuscript ‘Conversion of marginal lands into food production areas through an integrated green biotechnology approach: the case of mining wastes and Lablab purpureus (L.) Sweet.’ is noteworthy and fits into the scope of the Plants journal. The results presented are appropriate and thought-provoking, which generally deserve publication. The experimental procedure and analysis are mostly correct. However, the current version of the manuscript requires revisions and additions. I want to point out the issues I am concerned about.

pp. 168-170. The authors produced Technosols by mixing gossan wastes with a mix of inorganic and organic wastes from local industries (urban gardening services, quarries, cafes and breweries). What were the criteria for choosing these specific types of waste? They can be of highly different quality, even if comparing the organic parts of this spectrum.

pp. 249-252. The authors assessed the rehabilitation effectiveness of the Technosols containing gossan waste by measuring soil enzymatic activities as biological indicators, which reflect soil functional diversity, changes in microbial community composition, and microbial status. In this study, we see that Lablab influences the diversity of the microbial community. Soil enzymatic activity is indeed an informative indicator. However, rehabilitation is typically thought to be a managerial return for human use. Since the authors conclude that this case could be a reference model for the study of rehabilitation solutions for sulfide mining areas as the Iberian Pyrite Belt (pp. 524-525), it is highly recommended to include the description of common traits or typical characteristics of mine waste in the Iberian Pyrite Belt that are reflected in the studied gossan waste. Furthermore, it makes sense to consider material availability, covering both organic and inorganic substances to better understand the rehabilitation perspectives.

I honestly hope you will find my suggestions supportive.

Kind regards,

Reviewer

Author Response

Attached document

Reviewer 2 Report

This study aims to investigate the feasibility and effectiveness of biotechnology for mine waste (gossan wastes) rehabilitation using designed Technosol and a tolerant plant through a pot trial, possibly converting the marginal areas into food production zones. This study could serve as a reference for rehabilitating sulfide mining areas, such as the Iberian Pyrite Belt, particularly in harsh climates that may worsen with climate change, like the Mediterranean region. The study is relevant to the field and scientifically sound, with a clear experimental design and appropriate analyses of physicochemical, biological, and ecotoxicological characterisation, contributing to the practical solutions for mine waste rehabilitation.

However, there are several suggestions that should be handled seriously.

The Introduction section should be more concise and logical. There are many details which are not important to the main argument of this research. Some key components are missing, such as the background of related technologies (e.g., Technosols and phytostabilisation) used in this study, research gaps, questions, and hypothesis.

In the Results and Discussion section, the descriptions of results are very detailed, but the discussion and deeper understanding of the results related to key findings and arguments are not enough.

The Materials and Methods section lacks sample preparation methods for key analyses. The physical properties (structure and water holding capacity) can not be found in the physicochemical analysis.

In addition, the study will be much more robust if there are analyses such as the wet chemistry of porewater, the physical property of solid, mineralogy, forms of C, N, and P, and microbial community.

There are many tables in the manuscript that could be transformed into figures, if possible, to help readers interpret and understand the data and research.

The manuscript's language is generally good, but long sentences slow down the readers' understanding.

Author Response

Attached document

Reviewer 3 Report

see in-text (pdf) comments

Conversion of marginal lands into food production areas 2 through an integrated green biotechnology approach: the case 3 of mining wastes and Lablab purpureus (L.) Sweet. 4

Antonio Aguilar-Garrido 1,*, Marino Pedro Reyes-Martín 1, Patrícia Vidigal 2 and Maria Manuela Abreu 2

Important contribution in an series of publications from the Iberian Pyrite Belt.

Some minor language editing needed.

Self-referencing should be revised to the necessary: 18 (15%) references citing Abreu, 16 of which relate to the Sao Domingo site and touch issues covered in the presented work (technosols, phytoaccumulation, - stabilization, …)

Introduction

Since there has been so much published on the issue by at least one co-author (Abreu), the new aspects of the present paper have to be clearly highlighted and duplication of arguments of previous papers has to be reduced. What are the main outcomes of previous research of the authors (mini review) and what gaps of knowledge are solved with the paper presented (e.g., using Lablab instead of Cistus)?

34-62  Condense to the essential, much has been covered in previous publications

 63-83 Condense; clearer focus on the importance of post-mining sites as agricultural sites for coping w food security of a growing population in contrast to tox risks (that’s why most is reclaimed for forestry).

121 ff  specify secondary objectives as treated in the methods (to be done) and results-discussion section

Method section before results

Results

No need to repeat tables in text, point out essentials.

Conclusion

Any ideas for future research? Seems you can only speculate about long-term developments and corresponding effects on animals and food for humans from the rehabilitated sites? Can you be sure there will be no risk?

Author Response

Attached document

Reviewer 4 Report

Dear Authors,

As a whole, the manuscript appears not very strong and helpful for the scientific community. It is not well organized. In particular, you should put the Materials and methods section before the Results and discussion. Furthermore, the relevance of the research, the benefits in terms of soil restoration and environmental health aspects, should be better discussed.

The only part that appears adequate is the introduction section; however, it should also be improved.

Title: The title is too long.

Abstract, Lines 13-15: “three major problems: environmental contamination, management of post-mining, urban and food-industry wastes, and the growing food demand.”

Actually, I read more than 3 problems. Please, recheck it.

Abstract, general considerations: You should also provide quantitative data from your results.

Introduction section:

From line 35:The global mining industry is growing sharply…”

Please, provide quantitative data about the global mining industry, the waste that is generated, the number of active and abandoned mines and so on.

Lines 65-68: “Thus, to satisfy this growing number of consumers, production will also have to increase, mainly due to the expansion of land area, which will lead to a biodiversity loss and an increase in greenhouse gas emissions [18,19].”

Although you added something else in the subsequent lines, your statement is inaccurate and unbalanced. Please consider that many recent scientific works highlighted that food production should change to allow a lower land use, to reduce water consumption and greenhouse gas emissions. See for example:

-        Willett et al., 2019. Food in the Anthropocene: the EAT–Lancet Commission on healthy diets from sustainable food systems. The Lancet. DOI: https://doi.org/10.1016/S0140-6736(18)31788-4

-        James-Martin et al., 2022. Environmental sustainability in national food-based dietary guidelines: a global review. The Lancet Planetary Health. DOI: https://doi.org/10.1016/S2542-5196(22)00246-7

As a consequence, I recommend you to discuss the topic more deeply. It is currently being discussed too superficially.

Line 80: “Technosols”.

The first time that you introduce this term, you should explain it.

Line 95: “But the impact of some amendments may be weak in the long term [21,23]”

Why should it be weak in the long term? Please, explain it in the manuscript.

Results and discussion section: It is unusual to find Results and discussion section first, then the Materials and Methods. Please, put them in the conventional order. Indeed, the initial presence of the Materials and Methods section would make the manuscript easier to follow.

Lines 267-268: These results are contrary to what is found in the literature”.

Please, provide more details about the literature.

Materials and methods section: As previously mentioned, it is necessary to put this section before the Results and discussion. Other, more specific comments about the Materials and methods section are here below:

Experimental set-up sub-section: more specific details about the abandoned mining area of São Domingos are necessary. A map of the area would also be helpful.

Also, a more accurate description of gossan waste is necessary.

Furthermore, the time period of your analysis must be shared (i.e. when did you start the analysis? When did you finish them?).

English needs to be improved.

Author Response

Attached document

Reviewer 5 Report

I suggest to explain better the method. I suggest to characterize the amendments and describe the chemical concentration to understand better the remediation technology.

Why do you choose the natural amendments from waste and why that percentage. did you characterize the amendments?

There is an increase in As concentration, did you consider it is not a good results.Do you observed As 3+ or As5+,: As5+ can be immobilized by phosphate.

Why do you write increasing organic carbon there is an increase the mobility of other PHE, but non for Pb. Generally increasing OC you can observe a decreasing of mobility. 

Why do you consider pH=6.5 means not positive, did you test other pHs? Could you try to use lime  to change pH, increasing or decreasing the amount of lime?

What about CEC? When CEC decreases, PHE are not bound by clay particles in soils

Did you try different type of plants which could give you better results?

Where are data from Anova? Why didn't you published.

What about data from : Biological adsorption coefficient, Translocation coefficient , Soil-plant transfer coefficient?

Author Response

Attached document

Round 2

Reviewer 2 Report

Please check the Reference section after the manuscript is fully revised.

Author Response

Dear Reviewer 2, 

Thank you for your valuable comments on our manuscript ID plants-2431144 entitled “A green solution for the rehabilitation of marginal lands: the case of Lablab purpureus (L.) Sweet grown in Technosols.” to the Special Issue “Future Phytoremediation Practices for Metal-Contaminated Soils” of the journal Plants

When the manuscript was fully revised, we manually checked the reference section manually to make sure that everything was correct (e.g., consistency in abbreviated journal names, numerical order). 

Thank you for your time and consideration.

Yours sincerely,

the authors.

Reviewer 4 Report

Dear authors,

You improved your manuscript. I am glad about it.

However, I have the following recommendations:

  • Subsection 2.1. You wrote: “All substrata were incubated at 75% of water-holding capacity and room temperature for 30 days.” Please, clarify the room temperature.
  • Subsection 2.1 It is not clear to me which scientific procedure you followed. I mean, 30 days of incubation, 174 days after sowing etc. Is it a standardised procedure? If not, you must explain why you followed such a procedure.
  • Please, recheck the numerical order of your references.
  • Why did you define Pb, Hg and As “Potentially Hazardous Elements”? They are hazardous for elements for human health, not “potentially hazardous”.
  • The results from Table 2 seem very strange. You should adequately investigate and discuss possible reasons.

English needs to be improved.

Author Response

Attached document

Reviewer 5 Report

One point is not explained, why CEC is low after the treatment and the consequences due to this value.

If shoots are enriched with contaminants, why not to choose a different plant? why not wait some years before allowing animals to graze on this area?

Author Response

Attached document

Round 3

Reviewer 4 Report

The manuscript has been adequately improved. I am glad about it.

The manuscript has been adequately improved. I am glad about it.